# Simulations and active learning enable efficient identification of an experimentally-validated broad coronavirus inhibitor

Katarina Elez [1], Tim Hempel [1,2,3], Jonathan H. Shrimp [4], Nicole Moor[5,6], Lluís Raich[1], Cheila Rocha[5,6], Robin Winter[1,7], Tuan Le[1,7], Stefan Pöhlmann [5,6], Markus Hoffmann [5,6], Matthew D. Hall [4] & Frank Noé [1,2,3,8] ✉

Drug screening resembles finding a needle in a haystack: identifying a few effective inhibitors from a large pool of potential drugs. Large experimental screens are expensive and time-consuming, while virtual screening trades off computational efficiency and experimental correlation. Here we develop a framework that combines molecular dynamics (MD) simulations with active learning. Two components drastically reduce the number of candidates needing experimental testing to less than 20: (1) a target-specific score that evaluates target inhibition and (2) extensive MD simulations to generate a receptor ensemble. The active learning approach reduces the number of compounds requiring experimental testing to less than 10 and cuts computational costs by ∼29-fold. Using this framework, we discovered BMS-262084 as a potent inhibitor of TMPRSS2 (IC50 = 1.82 nM). Cell-based experiments confirmed BMS-262084's efficacy in blocking entry of various SARS-CoV-2 variants and other coronaviruses. The identified inhibitor holds promise for treating viral and other diseases involving TMPRSS2.

The efficient discovery of drugs is critical for the development of therapies against rapidly evolving diseases. Despite scientific advancements, drug discovery remains a slow and expensive process, characterized by high failure rates[1]. The initial phase of drug discovery, known as hit identification, is particularly challenging and can be regarded as a needle-in-a-haystack problem.

Virtual screening has enhanced the efficiency of exploring the chemical space, reducing the number of compounds that require experimental testing. However, relying on brute-force virtual screening might not solve the needle-in-a-haystack problem and is potentially very wasteful. Methods typically employed in virtual screening approaches include pharmacophore modeling[2], molecular docking[3], molecular dynamics (MD)[4] and machine learning (ML)[5]. While docking

methods and their associated scoring functions are highly efficient in screening through vast databases of candidate molecules, a heuristic for binding affinity they provide can be very inaccurate[6]. The quantity we are interested to predict is protein function, and how to inhibit that. Furthermore, many virtual screening approaches disregard that protein binding pockets can have multiple conformational states, which can play a crucial role[7], both in the context of induced fit and conformational selection mechanisms[8].

To explore the power of using target-specific information, protein flexibility and active learning in virtual screening, we focused on TMPRSS2, a human serine protease, whose inhibition mechanism was studied in detail[9]. TMPRSS2 is involved in prostate cancer[10], as well as cellular entry of influenza A, SARS-CoV and MERS-CoV viruses[11,12].

[1]Department of Mathematics and Computer Science, Freie Universität Berlin, Berlin, Germany. [2]Department of Physics, Freie Universität Berlin, Berlin, Germany. [3]Microsoft Research AI for Science, Berlin, Germany. [4]National Center for Advancing Translational Sciences, National Institutes of Health, Rockville, MD, USA. [5]Infection Biology Unit, German Primate Center - Leibniz Institute for Primate Research, Göttingen, Germany. [6]Faculty of Biology and Psychology, University Göttingen, Göttingen, Germany. [7]Department of Bioinformatics, Bayer AG, Berlin, Germany. [8]Department of Chemistry, Rice University, Houston, TX, USA. ✉e-mail: franknoe@microsoft.com

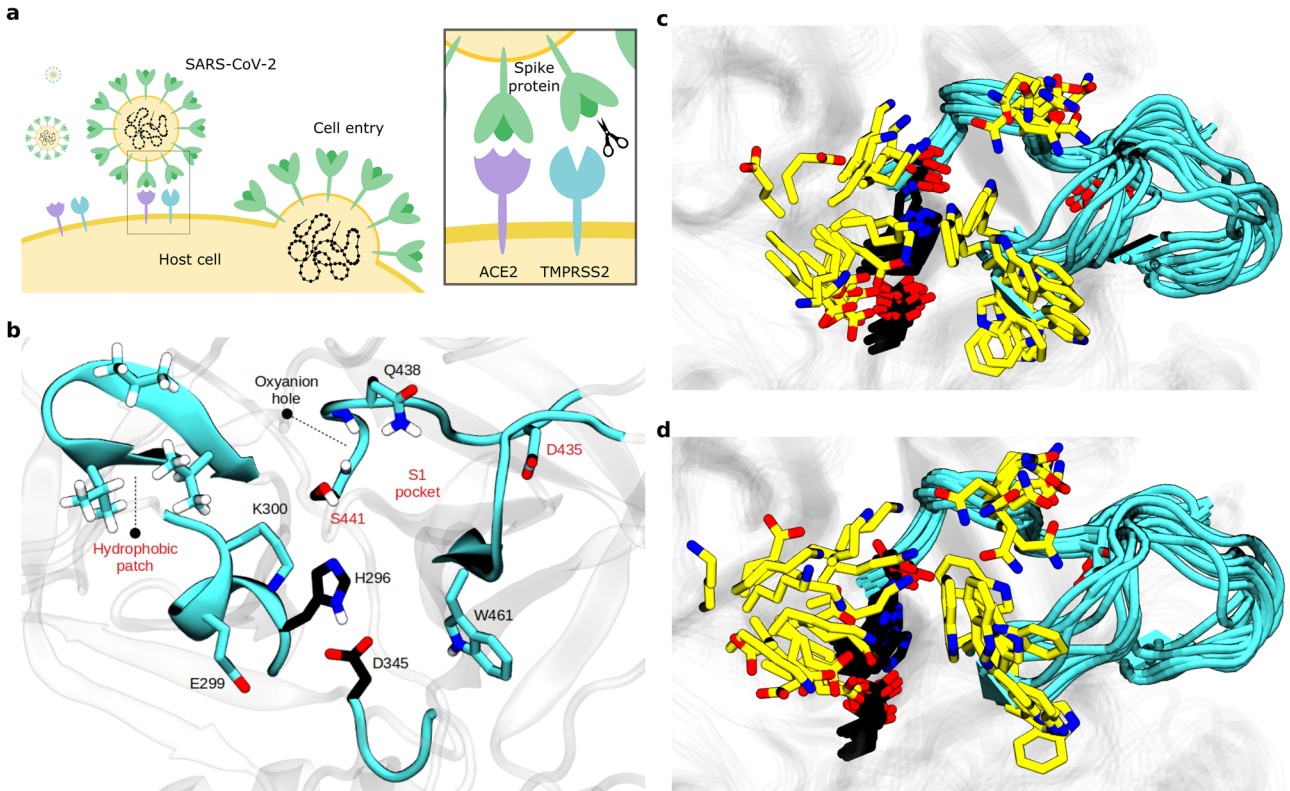

**Fig. 1 | Function and structure of TMPRSS2. a** TMPRSS2 role in Spike (S) protein-driven entry of SARS-CoV-2 into host cells. **b** Important structural features of TMPRSS2. Catalytic triad residues are depicted in black. Labels of the features included in our target-specific score are in red. **c**, **d** Structures belonging to receptor ensemble from apo and holo MD simulations.

Notably, it facilitates the entry of SARS-CoV-2[13] (Fig. 1a), a function that is retained for the Omicron variant[14,15]. Its known inhibitors either form a covalent bond with the enzymatic reactive center[16–19] (Fig. 1b) or establish a stable non-covalent complex[20].

In this manuscript, we present the development of an active learning approach for drug screening and apply it to TMPRSS2 inhibition. Our approach ranks candidates according to a target-specific score and efficiently navigates through chemical space in several cycles. Using our approach, we successfully identified BMS-262084, a potent nanomolar inhibitor of TMPRSS2 which effectively blocks coronavirus entry into Calu-3 human lung cells.

## Results

We develop an active learning approach for TMPRSS2 inhibition (Fig. 2a) and we apply it to the DrugBank library and the NCATS in-house library. The DrugBank contains four experimentally verified TMPRSS2 inhibitors: nafamostat, camostat, gabexate and otamixaban. To validate our approach, we simulate a virtual screen on the Drug-Bank library as follows: starting with 1% of the whole library, we employ an active learning cycle to select subsequent extension sets of the same size until the method scores all four known inhibitors. Our objective is that the known inhibitors receive high rankings – thus minimizing the number of experimental tests – while screening as few compounds as possible from the whole library – thus minimizing the computational cost. Subsequently, we use our approach to find TMPRSS2 inhibitors.

### Target-specific score enables hit discovery

We introduce an empirical score tailored to measure target inhibition, aiming to mitigate the inaccuracy of docking scores. An effective TMPRSS2 inhibitor (Fig. 1b), either physically occludes its active site (non-covalent inhibitor) or forms a stable enzyme-drug complex

(covalent inhibitor). Therefore, our proposed score (Eq. (1)) rewards the occlusion of the S1 pocket and the adjacent hydrophobic patch, as well as short distances for features that describe reactive and recognition states (see Target-specific scoring for details). In general, such a score can be learned (see Learned score generalizes to trypsin-domain proteins) when considering many protein targets and their corresponding inhibition data.

To evaluate our target-specific score (Fig. 2b, c), we use a dataset of compounds tested against TMPRSS2 from the NCATS OpenData Portal. Despite not being optimized for this dataset, our target-specific score computed from docking poses (static $h$-score, sensitivity of 0.5) outperforms the docking score (sensitivity of 0.38), emerging as a better model for serine protease inhibition. The number of false positives is likely underestimated as only the top 50 hits from the dataset were used and, therefore, we do not compute specificity values here.

Next, we compare the effectiveness of the docking score and our static $h$-score in the active learning cycle on the DrugBank library (Table 1, rows 1 and 2). We dock candidates to each of the 20 structures in our receptor ensemble and score the resulting docking poses. On average, the docking score requires computationally screening 2755.2 compounds (simulation time 15,612.8 h) compared to only 262.4 compounds (simulation time 1486.9 h) using the static $h$-score for ranking candidates. More importantly, using the docking score the four known inhibitors appear within the top 1299.4, whereas they are in the top 5.6 positions using the target-specific score, resulting in a more than 200-fold reduction in the number of compounds that need to be experimentally screened.

We examine structures where the docking score and the static $h$-score substantially differ (Fig. 3a) and find that the $h$-score effectively captures important structural features, distinguishing good inhibitors far more accurately than the docking score. The latter either

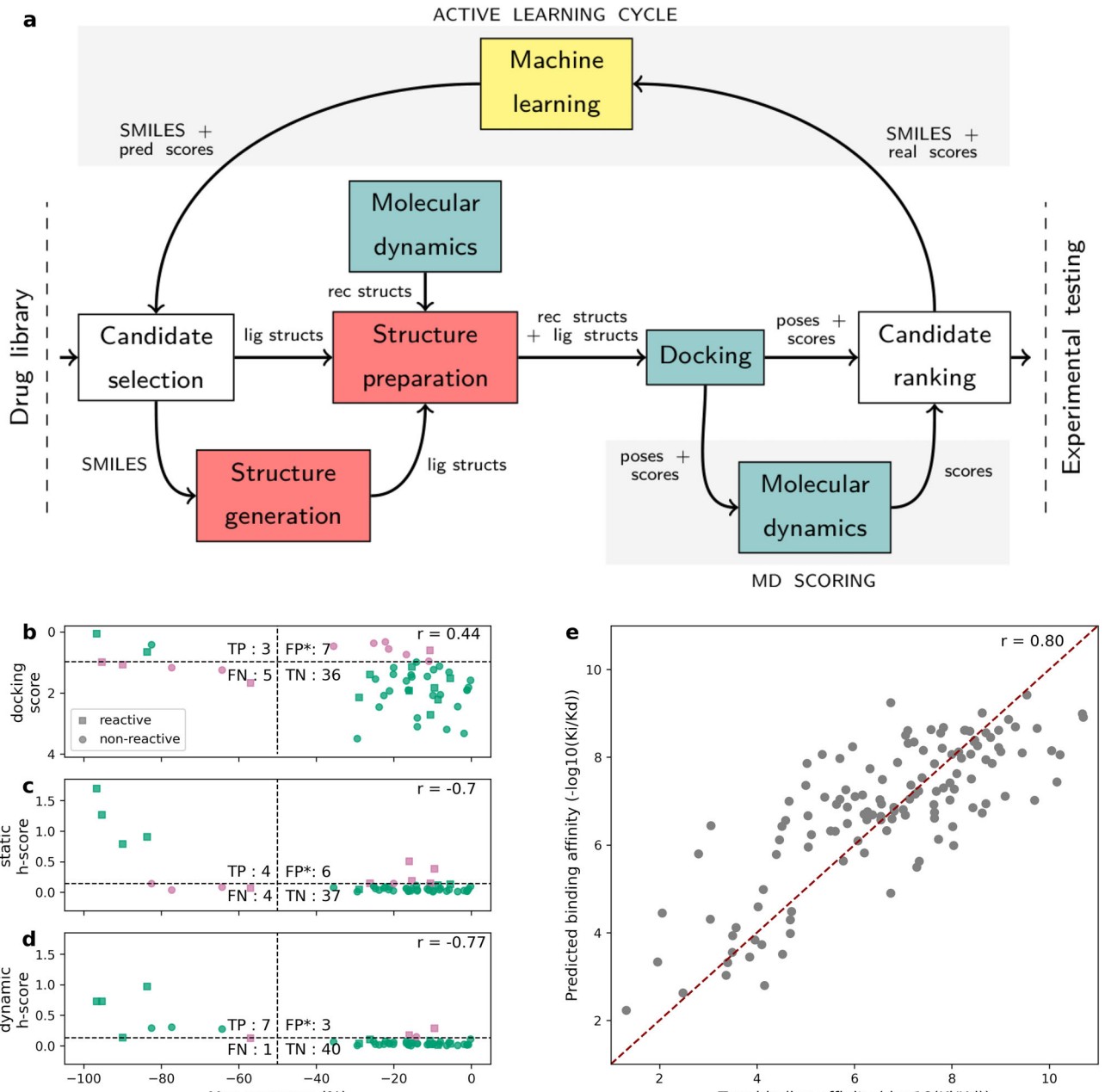

**Fig. 2 | Active learning approach, target-specific score and trypsin-domain-specific score. a** Schematic representation of our active learning approach. **b**–**d** Correlation between experimental drug efficacies (as measured by maximal response in the biochemical assay, x-axis) and different virtual screening scores (y-axis) - mean normalized *vinardo* docking score, h-score of the docking pose and dynamic h-score (averaged over MD trajectory), respectively. Green and pink denote correctly and incorrectly classified compounds, respectively. Squares mark potentially reactive compounds, circles non-reactive ones. **e** Correlation between experimental (true) binding affinities and predicted binding affinities for trypsin-domain proteins in PDBbind version 2020.

overestimates (e.g. nafamostat and otamixaban in Fig. 3c) or underestimates (e.g. camostat and gabexate in Fig. 3c) the inhibitory effect of a candidate.

## Learned score generalizes to trypsin-domain proteins

We hypothesize that our target-specific scoring method can be extended to other trypsin-domain proteins by learning a score based on the individual observables used in the *h*-score. Instead of relying on an empirical formulation that aggregates or selects specific features, we adopt a data-driven approach, training a model to predict binding affinities for proteins containing the trypsin domain using ΔSASA values and distances from the ligand for each residue in the S1 pocket and hydrophobic patch.

To evaluate this hypothesis, we select a subset of the PDBbind database containing experimental structures of trypsin-domain proteins bound to ligands, along with their binding affinity data. A simple random forest regressor, trained on the aforementioned observables for this subset, achieves a correlation of 0.80 between true and predicted binding affinities on the test set (Fig. 2e), demonstrating the learned scores ability to generalize to proteins containing the trypsin domain.

Feature importance analysis (Supplementary Fig. S1) identifies ΔSASA of the residue at the S1 pocket entrance (residue Trp461 in TMPRSS2) and the catalytic histidine as key predictive features, consistent with the expectation that strong binders shield this region. Additionally, the distance to the residue opposite the S1 pocket

**Table 1 | Comparison of different pipeline setups**

| Setup | Target | Ranking | Active learning | Compounds computationally screened | Total simulation time (h) | Compounds to screen experimentally | rs |
|---|---|---|---|---|---|---|---|
| Baseline | Receptor ensemble | Docking score | Yes | 2755.2 | *15,612.8 | 1299.4 | 0.4 |
| No MD scoring | Receptor ensemble | Static h-score | Yes | **262.4** | *1486.9 | 5.6 | 0.2 |
| Full active learning approach | Receptor ensemble | Dynamic h-score | Yes | **246.0** | *3403.0 | 7.6 | **1.0** |
| No receptor ensemble | Homology model | Dynamic h-score | Yes | 754.4 | **829.8** | 709.0 | 0.0 |
| No receptor ensemble nor MD scoring | Homology model | Static h-score | Yes | 2230.4 | **631.9** | 2179.8 | 0.6 |
| No active learning cycle | Receptor ensemble | Dynamic h-score | No | 7166.8 | *99,140.7 | 16.6 | **1.0** |

Target structure, ranking method, usage of active learning, number of computationally screened compounds, total simulation (docking + molecular dynamics) time, number of compounds to screen experimentally and Spearman's coefficient for rank correlation of four known inhibitors for different pipeline setups. Average (mean) values over five replicates are reported. Total simulation time is expressed in core hours, which indicates one CPU/GPU being used for one hour of execution time. Total simulation times denoted by * additionally include the time required to generate the receptor ensemble, which we estimate to be ≈8000 h. Note that Spearman's coefficient was calculated using only four known inhibitors and this limited sample size may influence the statistical reliability of the correlation. The best results in each category are marked in bold.

(residue Lys300 in TMPRSS2) emerges as an important factor, supporting the idea that potent inhibitors extend toward this patch.

We recommend using the learned score for future investigations of trypsin-domain proteins. For the subsequent computational and experimental screens, we now resort to the TMRPSS2-specific score.

### Molecular dynamics allow accurate candidate ranking and docking

We also conduct experiments to evaluate the significance of MD simulations within our active learning approach. MD was used in two ways: (1) to generate 10-ns simulations of protein-ligand complexes for dynamic h-scoring, totaling 100 ns per ligand and 818 μs for all 8180 ligands and (2) to generate a ≈100-μs simulation of the receptor from which 20 snapshots are used for docking ("receptor ensemble", Fig. 1c, d).

We first examine the relevance of running MD simulations for inhibitor scoring. MD seeded from docked poses can reduce false positives/negatives by expelling a misposed ligand from the active site or relaxing a non-optimal pose. Indeed, computing the score from MD simulations (dynamic h-score) further improves the classification of TMPRSS2 inhibitors from the NCATS OpenData Portal (Fig. 2c, d), increasing sensitivity to 0.88.

How relevant is MD-based scoring for finding the four known inhibitors in DrugBank? The number of compounds requiring computational and experimental screening is similar between static and dynamic h-scoring (Table 1, rows 2 and 3), suggesting that the dynamic h-score does not provide a significant benefit in our case while doubling the computational cost. However, it increases the correlation of the known inhibitors' rankings from 0.2 to 1.0, which indicates that dynamic scoring may be more robust and beneficial with other targets.

Next, we evaluate the effect of dynamics on the target by removing the MD-generated receptor ensemble. Instead, we dock candidates to a single homology model and rank them by their dynamic h-score. This increases the average number of computationally screened compounds to 754.4 (simulation time 829.8 h) and, more importantly, results in poor ranking of the known inhibitors (within the top 709.0 compounds). This result underscores the importance of having a receptor ensemble, which increases the likelihood of docking to binding-competent target structures.

Finally, we remove both the receptor ensemble and MD scoring, docking candidates to a single homology model and ranking them by their static h-score. This substantially increases the average number of compounds screened to 2230.4 (simulation time 631.9 h) and produces an almost useless ranking, emphasizing the critical role of MD in achieving meaningful results.

To better understand the role of MD from a structural perspective, we compare the poses of known inhibitors docked to different target structures. Docking to one of the MD-generated receptors (example of a good structure) produces high-scoring poses while docking to the homology model (example of a bad structure) results in consistently low scores, using both static and dynamic h-score (Fig. 3b, c). Moreover, MD scoring can correct docking artifacts when a candidate docks into a sub-optimal pose (e.g. camostat in Fig. 3b) or an unstable high-scoring pose (e.g. otamixaban in Fig. 3b).

### Active learning vastly accelerates compound search

Even though our dynamic target-specific score is effective for ranking drug candidates, it requires expensive MD simulations, which may not be feasible for screening large compound libraries. To tackle this problem, we use the active learning cycle (see Active learning cycle).

To explore the effect of the active learning cycle on the computational burden, we simulate another virtual screen where candidates are selected randomly. Without an active learning cycle, random selection requires screening 7166.8 compounds (simulation time 99,140.7 h) and the known inhibitors appear within the top 16.6

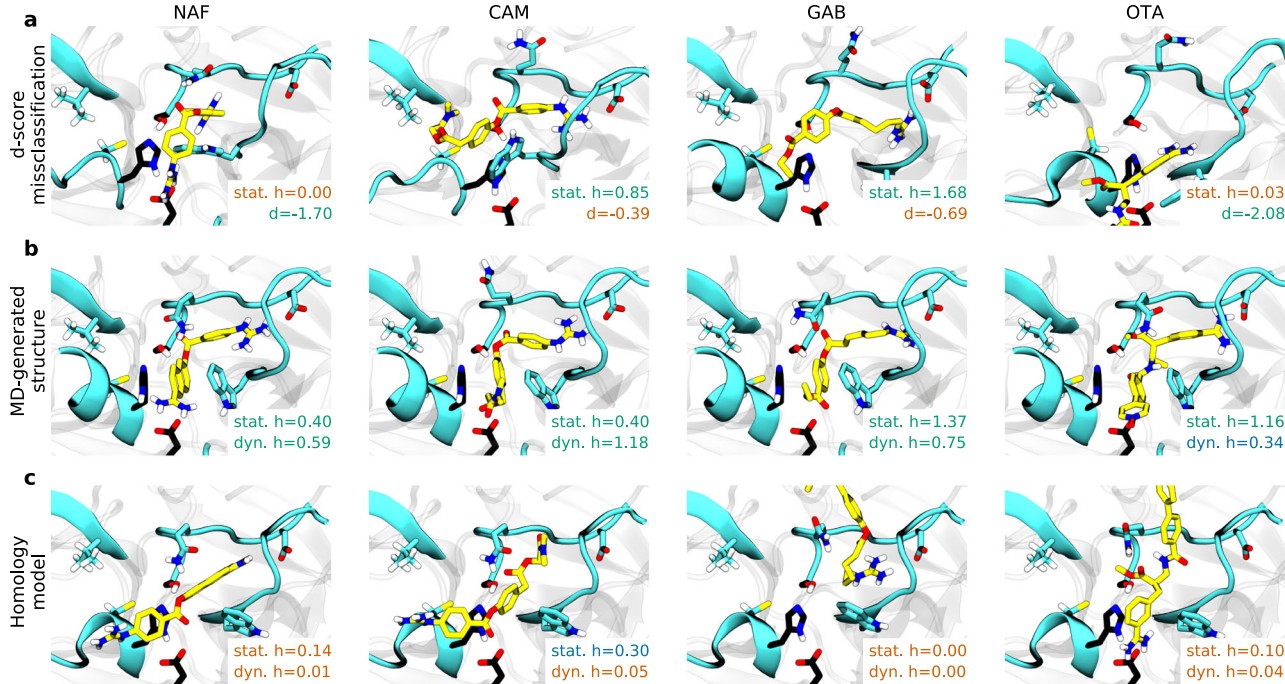

**Fig. 3 | Effect of target-specific score and molecular dynamics on candidate scoring. a** Examples of docking poses of known inhibitors misclassified by docking score with respective static *h*-score and normalized docking score. Scores are color coded as green for good, blue for intermediate and orange for bad. **b, c** Examples of docking poses of known inhibitors for good (MD-generated) vs bad (homology model) receptor structure with respective static and dynamic *h*-scores.

positions on average. Therefore, using a machine learning model to select extension sets of candidates reduces the computational burden by ~29-fold.

### Active learning approach identifies inhibitors of TMPRSS2

To identify compounds with promising inhibitory properties against TMPRSS2, we examined the predictions of our active learning approach on the DrugBank after screening 10% of the whole library (Supplementary Table S1).

The best-ranked compound, DB03417, was characterized by a high *h*-score and showed potential upon visual inspection. This compound maps to a crystal structure (PDB ID: 1RXP[21]) in which its parent compound (chemical relation in Supplementary Fig. S2), known as BMS-262084, engages in covalent binding with trypsin. Leveraging this information, we moved forward with scoring BMS-262084 (*h*-score = 1.249) and assessing its inhibitory potential.

We experimentally evaluated the effect of BMS262084 in a TMPRSS2 biochemical assay (Fig. 4a). The inhibitory profile of BMS-262084 IC50 = 1.82 nM was better than that of camostat IC50 = 3.17 nM and comparable to that of nafamostat IC50 = 1.08 nM, the most potent known inhibitor of TMPRSS2.

Next, we repeated the same experiment with different pre-incubation times to understand the time-dependence of inhibitor potencies. Across pre-incubation times of 1, 4 and 8 h, the three inhibitors showed similar inhibitory profiles to those of the initial experiment (Fig. 4b). However, from the 18 h pre-incubation mark, inhibitor potencies decreased. This decline was more pronounced for camostat and nafamostat that converged to the same IC50 after 48 h of pre-incubation. In contrast, BMS-262084 showed a 5-fold higher IC50 for the same pre-incubation duration.

We also applied our active learning approach to the NCATS in-house library containing ~145,000 compounds. In the first round, we used the docking score to select 1100 compounds for experimental validation, while in the second round, we used the dynamic *h*-score to select an additional 500.

Experimental validation of our predictions on the NCATS in-house library revealed 33 compounds with a maximum response below −40% (molecular graphs in Supplementary Figs. S3 and S4). Among these, otamixaban (IC50 = 0.79 μM), dabigatran ethyl ester (IC50 = 2.24 μM) and two more compounds (IC50 = 8.91 μM for both) exhibited an IC50 below 10 μM, representing promising scaffolds for further optimization.

### BMS-262084 blocks coronavirus entry into Calu-3 cells

We investigated the impact of our most potent inhibitor, BMS-262084, on cell entry of live SARS-CoV-2 and pseudovirus particles bearing coronavirus S proteins into Calu-3 human lung cells (TMPRSS2-positive).

We first analyzed the inhibitory effect of BMS-262084 in the context of the live SARS-CoV-2 virus (Supplementary Fig. S5). Pre-incubation of Calu-3 cells with BMS-262084 at noncytotoxic concentrations (Supplementary Fig. S6) strongly inhibited the relative (compared to no inhibitor) infectivity of SARS-CoV-2 with an IC50 of 0.51 μM, as evidenced by a reduction in SARS-CoV-2 nucleoprotein signals in infected cells at 24 h postinoculation (Supplementary Fig. S7). We also conducted live-virus inhibition experiments comparing BMS-262084 with camostat (Fig. 4c, d and Supplementary Fig. S8). BMS-262084 showed greater efficacy against both AY.1 (Delta) and KP.3.1.1 (recent Omicron sublineage), with IC50 values of 8.66 nM and 8.03 nM, respectively, making it more potent than camostat (22.05 nM and 38.30 nM) in blocking infection of Calu-3 lung cells.

Next, we preincubated Calu-3 cells with different concentrations of BMS-262084 or camostat before adding pseudoviruses carrying diverse coronavirus S proteins. This included S proteins of four SARS-CoV-2 lineages - B.1 (early pandemic), B.1.617.2 (Delta variant), EG.5.1 (XBB-sublineage of Omicron variant, circulating in 2023), BA.2.86 (Omicron subvariant, dominating lineage in 2024). In addition, we analyzed S-proteins of four additional coronaviruses that can infect humans: HCoV-NL63 and HCoV-229E, which are seasonal coronaviruses causing common cold, as well as the zoonotic coronavirus

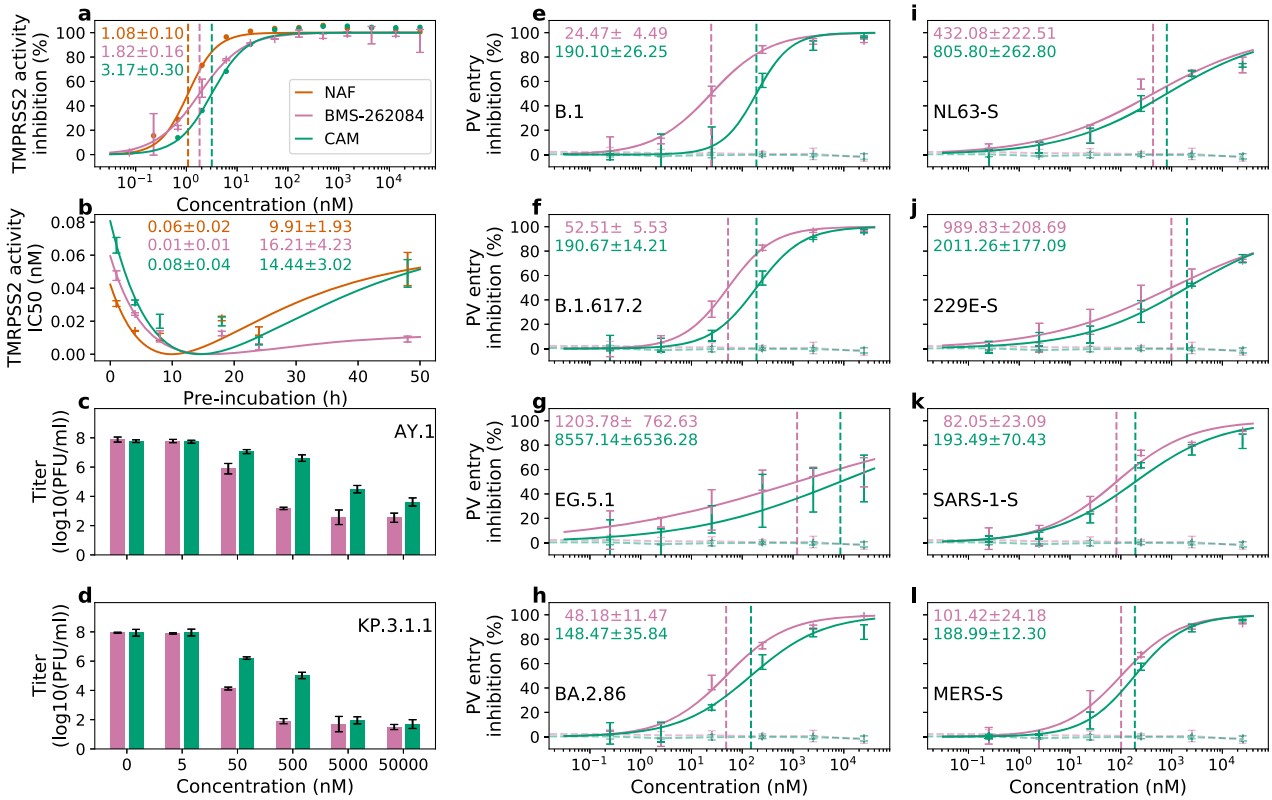

**Fig. 4 | Experimental results of BMS-262084 inhibition. a** Dose-response curves and IC50 estimates for inhibition in TMPRSS2 biochemical assay. For BMS-262084, the average (mean) ±SD of three technical replicates is shown. **b** IC50 as a function of pre-incubation time, estimates of IC50 at infinite time and time at which the function reaches its minimum. **c, d** Inhibition of live SARS-CoV-2 infection of Calu-3 cells. PFU, plaque-forming units. The average (mean) ±SD of three technical replicates is shown. Statistical significance was analyzed by two-way analysis of variance (ANOVA) with Dunnetts post hoc test. $P$ values (for concentrations between 5 and 50,000 nM, from left to right) are as follows: AY.1 (0.9911, 0.9992, <0.0001, 0.0031,

<0.0001, <0.0001, <0.0001, <0.0001, <0.0001, <0.0001) and KP.3.1.1 (0.9995, 1.0000, <0.0001, <0.0001, <0.0001, <0.0001, <0.0001, <0.0001, <0.0001). **e–l** Dose-response curves and IC50 estimates for inhibition of pseudovirus cell entry into Calu-3 cells driven by VSV-G (control, dashed lines) or S protein (solid lines) of SARS-CoV-2 lineages B.1, B.1.617.2, EG.5.1 and BA.2.86 or coronaviruses HCoV-NL63, HCoV-229E, SARS-CoV-1 and MERS-CoV, respectively. The average (mean) ±SD of three biological replicates is shown. Each biological replicate was performed with four technical replicates.

SARS-CoV-1 and MERS-CoV, which can cause life-threatening disease in humans.

For Calu-3 cell entry of particles bearing either B.1-S, B.1.617.2-S or BA.2.86-S, similar inhibition profiles were observed for the two inhibitors (Fig. 4e, f, h), with strong inhibition by BMS-262084 (IC50 = 24.47 nM, IC50 = 52.51 nM and IC50 = 48.18 nM for B.1, B.1.617.2 and BA.2.86, respectively) and by camostat (IC50 = 190.10 nM, IC50 = 190.67 nM and IC50 = 148.47 nM for B.1, B.1.617.2 and BA.2.86, respectively). Inhibition of EG.5.1-S protein-driven Calu-3 cell entry by BMS-262084 IC50 = 1.20 μM or camostat IC50 = 8.56 μM (Fig. 4g) was ~20–50-fold less efficient compared to particles bearing B.1-S, B.1.617.2-S or BA.2.86-S. Even though both BMS-262084 and camostat showed a robust inhibition profile, BMS-262084 was ~3–8-fold more potent than camostat. In addition, for inhibiting Calu-3 cell entry of particles bearing either NL63-S, 229E-S, SARS-1-S or MERS-S, BMS-262084 was consistently ~2-fold more potent than camostat (Fig. 4i–l).

**Structural basis of TMPRSS2 inhibition by BMS-262084**

Based on the general mechanism of action of β-lactam inhibitors, we propose a mechanism of action for BMS-262084, which includes an initial binding step and two subsequent reaction steps (Fig. 5a). Upon binding of BMS-262084 to TMPRSS2, a non-covalent substrate enzyme complex is formed. In the first reaction step (acylation), the catalytic histidine (His296) deprotonates the catalytic serine (Ser441), which attacks the carbonyl center of the β-lactam ring to establish an acyl-

enzyme intermediate. In the second reaction step (hydrolysis), the catalytic histidine activates an incoming water molecule, which attacks the acyl-enzyme intermediate, releasing the hydrolyzed substrate and reinstating TMPRSS2 in its active form.

Using Markov state modeling, we analyze 40 MD simulations of 1 μs each and identify three metastable states of BMS-262084 binding to TMPRSS2 (Supplementary Fig. S9). In all these, BMS-262084's head binds to the S1 pocket of TMPRSS2. The three states differ in the position of the β-lactam ring and, consequently, in the orientation of BMS-262084's tail, which either points toward the hydrophobic patch (state 1) or away from it (states 2 and 3).

BMS-262084 binds to TMPRSS2 (Fig. 5b), with its guanidinobutane group forming a typical salt bridge to the aspartate at the bottom of the S1 pocket (Asp435). The strong interaction between the positively charged guanidine moiety and the negatively charged carboxylate of the aspartate is a known recognition mechanism, also exploited by nafamostat and camostat, both featuring a guanidinobenzoyl head instead of the BMS-262084's guanidinobutane.

The first metastable state (Fig. 5b) is characterized by the β-lactam ring of BMS-262084 positioned atop the catalytic serine (Ser441). We consider this conformation reactive when the β-lactam ring's carbonyl center is sufficiently close to the oxygen of the serine, enabling a suitable configuration for a nucleophilic attack. In our simulations, this reactive configuration is rarely observed. Opposite the S1 pocket, BMS-262084's formylpiperazine extends over the Cys281-Cys297 disulfide bridge, with

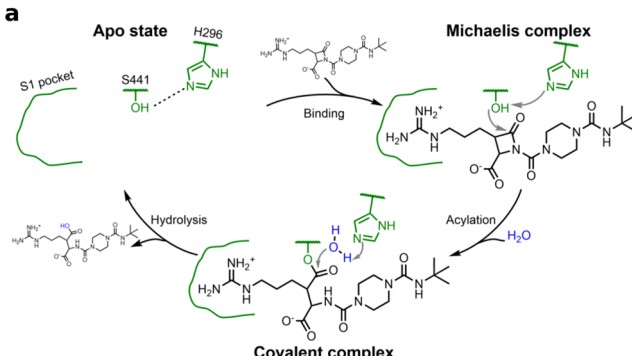

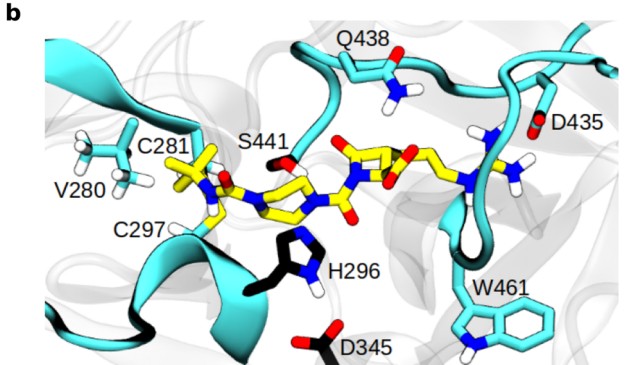

**Fig. 5 | Mechanism of action and binding mode of BMS-262084. a** Expected mechanism of action of $\beta$-lactam inhibitors applied to BMS-262084 inhibition of TMPRSS2. Relevant parts of the enzyme are depicted in green. **b** Binding mode of BMS-262084 to TMPRSS2 from MD simulations. Catalytic triad residues are depicted in black.

its hydrophobic N-tert-butylformamide interacting with Val280 from the hydrophobic patch of TMPRSS2. This conformation resembles the crystal structure of BMS-262084 in a covalent complex with bovine trypsin (Supplementary Fig. S10). Alternatively, the tail of BMS-262084 can be found in contact with the catalytic histidine (His296).

## Discussion

Targeting TMPRSS2 is significant for developing therapeutics against respiratory viruses, given its integral role in facilitating the cell entry of influenza A[11], coronaviruses[12] and certain paramyxo-(i.e. parainfluenza virus)[22] and pneumoviruses (i.e. metapneumovirus)[23]. Here, we developed an active learning approach for drug screening and applied it to TMPRSS2 inhibition, discovering BMS-262084 as a potent inhibitor.

We introduced a simple target-specific score that addressed the docking score limitations. Remarkably, the $h$-score was capable of identifying at least three active compounds (nafamostat, camostat and DB03417/BMS-262084) within the top 5 of the entire DrugBank library, showcasing its usefulness in hit discovery.

The static $h$-score demonstrated clear advantages over the conventional docking score, offering increased accuracy with minimal computational expenses. The learned version of the static $h$-score generalizes effectively to a broader family of trypsin-domain proteins, providing a framework for hit discovery across serine protease targets that are similar to TMPRSS2.

While the dynamic $h$-score delivers excellent performance, its reliance on MD simulations incurs higher costs, albeit lower than popular free energy calculation methods[24]. To balance computational cost and predictive accuracy, we suggest employing the static $h$-score as an initial filter, followed by MD simulations and dynamic $h$-score calculation for a subset of the most promising compounds.

Our approach enabled a direct comparison between static and dynamic methods using the same scoring metric, allowing for a clearer evaluation of their respective strengths. Based on our insights, we offer a strategy for developing static/dynamic mechanism-informed scores and recommend investing computational resources in running MD simulations to generate receptor ensembles that better capture target flexibility. We also provide a unified scoring framework that applies to both covalent and noncovalent inhibitors, offering an effective and simpler alternative to conventional covalent docking tools.

We demonstrate the scalability of our approach by successfully applying it to the NCATS in-house library of ~145,000 compounds. This screening effort identified otamixaban which we further investigated in combination with nafamostat or camostat in a separate study[20].

Recent works have shown that TMPRSS2 remains an important host cell factor for lung cell entry of contemporary SARS-CoV-2 lineages[14,25], especially in more relevant models such as human airway and intestinal organoids[15]. In line with those studies, our experimental validation robustly affirmed BMS-262084's effectiveness in inhibiting SARS-CoV-2 entry across multiple lineages, including two descendants of the Omicron variant.

Although BMS-262084 and camostat were both able to inhibit S protein-driven Calu-3 cell entry, BMS-262084 did so with superior efficiency as indicated by its ~3–8-fold and ~2-fold lower IC50 values for all SARS-CoV-2 lineages and coronaviruses tested, respectively. Moreover, it showed a more robust inhibitory profile at longer pre-incubation times. Our findings confirm BMS-262084's relevance and encourage further research on this compound to explore its suitability as an antiviral therapeutic.

The identification of a nanomolar inhibitor through virtual screening, as presented in this study, remains a notable accomplishment in computer-aided drug discovery[1]. Remarkably, our method may extend beyond the specific case of TMPRSS2, presenting a valuable tool for discovering inhibitors targeting other serine proteases. In addition, it shows promise for broader applicability across various systems, provided that their underlying mechanism is understood.

## Methods

### Target structures

At the time this study was conducted, there was no experimentally determined structure of TMPRSS2 and we use models of the catalytic domain from ref. 26 instead. Specifically, we exploit a homology model based on Enteropeptidase-1 (PDB: 3W94) in apo and holo conformations, the latter being docked complexes of TMPRSS2 with known inhibitors camostat[16] or nafamostat[17]. Our constructs contain residues 256–489 or residues 256–491, for apo and holo structures, respectively.

To eliminate possible artifacts of the models[27] and to allow for target flexibility we conduct molecular dynamics (MD) simulations of both apo and holo structures. We run the simulations with OpenMM 7.4.0[28] using the CHARMM 36 force field (version from 2019)[29]. Further details of the setup can be found in ref. 9.

We use hidden Markov models[30] on the resulting datasets to select representative target conformations. Specifically, we pick a total of 20 target structures – 10 from apo simulations, 4 from nafamostat-bound and 6 from camostat-bound simulations. For selecting structures, we use three features to account for S1-pocket entrance-loop openness, occlusion of S1-pocket by Trp461 and occlusion of Asp435 by charged residues.

### Drug libraries

We obtained the publicly available TMPRSS2 enzymatic activity screen from the NCATS Open Data Portal[31] (downloaded on October 29, 2020). The top 50 compounds were selected according to AUC.

We download the DrugBank database (version 5.1.6)[32] and extract all small molecules that have a specified SMILES string (10,758 compounds). We then filter out: (1) salts, (2) compounds with a molecular weight above 550 Da, (3) compounds with less than 6 heavy atoms and (4) compounds containing transition or post-transition metals. The resulting library consists of 8918 compounds.

Several in-house libraries (Genesis, Sytravon, NPACT and NPC) of the National Center for Advancing Translational Sciences (NCATS) are readily available for high-throughput screening. These libraries together amount to a total of ~145,000 compounds that we refer to as the NCATS in-house library.

## Docking and scoring

To obtain the three-dimensional structure of each ligand for docking, we either retrieve it from the ZINC database[33], in case a reference molecule is available, or we generate it from the SMILES string with the optimal ionization states at pH 7.05 using LigPrep from the Schrödinger Suite 2020-2. We prepare all receptor and ligand structures with MGLTools 1.5.6[34].

We dock each ligand structure against each of the 20 receptor structures using a fork of AutoDock Vina[35] called smina (version 2017.11.9)[36]. Docking is performed in a search space of size 30 Å centered around the catalytic serine (Ser441). We set the exhaustiveness to 10 and use the Vinardo scoring function[37] which is suitable for virtual screening purposes. Five residues are kept flexible throughout the docking run, namely: Glu299, Lys300, Asp435, Gln438 and Trp461. For each receptor-ligand pair, we only keep the best pose. The docked complex is finally assembled by combining: (a) the rigid part of the receptor, (b) the flexible part of the receptor associated with the best pose and (c) the best pose of the ligand.

To get the score for each ligand we first normalize the raw docking scores per receptor and filter out poses in which the ligand does not form at least 2 contacts (based on heavy atom distance and a threshold of 3.5 Å) with the S1 pocket (residues 435–441 and 459–464). We then compute the mean of the normalized docking score (of retained receptor-ligand poses) for each ligand.

## Target-specific scoring

We propose a score that summarizes our insights about the mechanism of action of TMPRSS2 inhibitors[9], termed the $h$-score:

$$h = \frac{\Delta SASA(S1)\Delta SASA(H)}{d^2(\text{react})d^2(\text{recog})n^{\frac{4}{3}}} \qquad (1)$$

where S1 is the S1 pocket (residues 435–441 and 459–464), H is the adjacent hydrophobic patch (residues 279–281 and 296–300), $d(\text{react})$ is the distance between the closest cleavable bond of the ligand and the oxygen of the catalytic serine (Ser441), $d(\text{recog})$ is the minimal distance between the ligand and heavy atoms of Asp435 (major substrate recognition residue at the bottom of the S1 pocket). The factor $n^{-4/3}$ compensates for the bias towards large molecules otherwise posed by the two SASA differences.

The observables for the $h$-score (Eq. (1)) are computed using MDTraj 1.9.4[38]. The difference in solvent accessible surface area upon binding ($\Delta SASA$) is computed with MDTraj's implementation of the Shrake–Rupley algorithm[39].

Possible cleavable bonds of the ligand are detected using RDKit 2020.03.4 and SMARTS patterns for the following classes of compounds: ester, phenylmethylsulfonyl fluoride (PMSF), chloromethyl ketone (CMK), aldehyde, trifluoromethyl ketone (TFK) and $\beta$-lactam. If a ligand does not contain a cleavable bond and, therefore, cannot react with the protease, we set this distance to its average value of 0.827 nm to not bias non-covalent compounds versus covalent ones.

The static $h$-score is computed as the mean of the top 3 equilibrated docking poses, each from a different receptor structure, to rule

out docking artifacts, and the dynamic score is calculated as the mean of the top 3 trajectories, each from a different receptor structure, over all frames of the MD simulation (in 1 ns steps).

## Trypsin-domain-specific scoring

We propose a learned version of the $h$-score that generalizes to targets containing the trypsin domain. To construct the dataset, PDBbind version 2020 was downloaded and annotated using the InterPro database. Entries with reported binding affinities as exact Ki or Kd values and the annotation IPR001254 (Serine proteases, trypsin domain) were retained. Receptor structures were repaired using PDBFixer to replace non-standard amino acids and add missing heavy atoms, followed by removal of inactive chains (with no residues within 5 Å of the ligand).

Pairwise alignments of the receptors were generated using TMalign[40] and receptors with a high mean pairwise distance (above 0.6) were excluded. The pairwise distances were used to conduct hierarchical clustering with SciPy's average linkage method. Closely related structures were grouped and progressively aligned until all structures were unified into a single aligned group. A representative structure was selected based on the lowest sum of distances to all other receptors. We manually annotated its S1 pocket and hydrophobic patch residues considering the TMPRSS2 structure. A residue correspondence between the representative and each receptor was established using the Needleman–Wunsch algorithm[41] on the positions of Cα atoms of the superposed structures. Receptors with missing S1 pocket or hydrophobic patch residues, or mutated catalytic serine, were excluded.

We calculated ΔSASA and distance from ligand for each S1 pocket and hydrophobic patch residue. Complexes in which the ligand was not bound near the S1 pocket (ΔSASA(S1) below 0.5) were excluded. The resulting dataset of 651 complexes was stratified based on the distribution of binding affinities, which were divided into 20 percentile-based bins, and split into 80% training and 20% test sets. A random forest regressor with 200 estimators was then trained on these observables to predict -$log$10(Ki/Kd).

## Molecular dynamics simulations

We use the AMBER ff14SB force field[42] for the receptor and the openff-1.1.0 small molecule force field[43] for the ligand, which is parameterized from the SMILES string with the openff toolkit (version 0.7.1)[44] and the openmmforcefields package (version 0.8.0).

The setup and subsequent production runs are carried out with OpenMM 7.4.0[28] in a cubic periodic box of 7.2 nm side length with TIP3P water[45] and a 0.1 mol/L NaCl ion concentration (neutral charge).

Molecular dynamics simulations are automatically seeded from the docking pose of the receptor-ligand pair. The solvent is generated for every receptor-ligand pair individually. It is equilibrated with constraints on the heavy atoms of receptor and ligand for 0.1 ns in the NVT ensemble and, subsequently, for 0.9 ns in the NPT ensemble at 310 K (physiological temperature) and 1 bar. We choose a Langevin integrator with a time step of 2 fs at the equilibration phase. In the production phase, we apply hydrogen mass repartitioning[46] and a 4 fs integration step with hydrogen bond restraints.

## Markov state modeling

We analyzed MD simulations by employing PyEMMA 2.5.7[47] to calculate inverse minimal distances between protein residues and various drug groups (guanidinobutane, beta-lactam ring, carboxylate, formylpiperazine, and tert-butylformamide). We conduct a linear VAMP[48] dimension reduction operation with a 5 ns lag time, utilizing the top 8 dimensions with the highest kinetic variance. Next, we perform k-means clustering with 180 cluster centers and estimate a 4-state hidden Markov model (HMM)[30] at a lag time of 1 ns, enabling the assignment of metastable states (binding modes).

## Active learning cycle

We define an active learning cycle that iteratively trains a machine learning (ML) model on the already screened candidates and selects new candidates for screening. To this end, it takes advantage of a pretrained deep learning autoencoder[49] to encode the SMILES of already screened candidates in a continuous latent space. These continuous and data-driven descriptors (CDDD) encodings are then used together with the associated scores (mean normalized docking scores or *h*-scores) to train a support vector regressor (SVR) using scikit-learn 0.22.1[50]. Once trained, this model predicts the scores on the subset of the library that has not been screened yet and selects candidates for the next round. We repeat this procedure for a certain number of rounds. The steps for screening the initial set of candidates and the subsequent extension sets are detailed below, while the algorithm is available in the SI (Alg. S1).

Initial set screening steps are:

1. Get Morgan fingerprints for the whole library;
2. Cluster Morgan fingerprints into *n_clusters* clusters using the *k*-means algorithm with 20 initializations and a maximum of 400 iterations;
3. Get a diverse initial set of candidates by taking one representative from each cluster;
4. Score the initial set of candidates.

Extension set screening steps are:

1. Get the set of potential candidates by subtracting the screened set of candidates from the whole library;
2. Train a support vector regressor (SVR) with default parameters on CDDD encodings and scores of screened candidates;
3. Predict scores for potential candidates from their CDDD encodings;
4. Get the extension set of candidates by taking top *ext_size* candidates with the best predicted score;
5. Score the extension set of candidates.

## TMPRSS2 biochemical assay

Inhibitors of TMPRSS2 were tested using a high throughput activity assay. The experiment was performed in a 1536-well black plate according to the published protocol[51]: Boc-Gln-Ala-Arg-AMC substrate (20 nl) and inhibitor (20 nl in DMSO) were added using an ECHO 655 acoustic dispenser (LabCyte). TMPRSS2 (5 µl, 0.018 mg/ml) in assay buffer (50 mM Tris pH 8, 150 mM NaCl, 0.01% Tween20) was dispensed to that, using a BioRAPTR (Beckman Coulter), for a total assay volume of 5 µl.

For pre-incubation: To a 1536-well black plate was added DMSO (20 nl) and inhibitor (20 nl, 250×) using an ECHO 655 acoustic dispenser (LabCyte). To that was dispensed TMPRSS2 (5 µl, 0.018 mg/ml) in assay buffer (50 mM Tris pH 8, 150 mM NaCl, 0.01% Tween20) using a BioRAPTR (Beckman Coulter) to give a total reaction volume of 5 µl. Following the desired pre-incubation time of the inhibitor with TMPRSS2, the fluorogenic peptide substrate, Boc-Gln-Ala-Arg-AMC, was added at 20 nl (10 µM, 250×). The final assay conditions are 10 µM peptide, 0.018 mg/ml TMPRSS2 in assay buffer (50 mM Tris-HCl, pH 8, 150 mM NaCl, 0.01% Tween20).

After incubation at room temperature for 1 h, fluorescence was measured. PHERAstar with excitation at 340 nm and emission at 440 nm was used for detection. Raw plate reads for each titration point were normalized relative to a positive control containing no enzyme (0% activity, full inhibition) and a negative control containing DMSO-only wells (100% activity, basal activity). Data normalization was performed using GraphPad Prism (GraphPad Software, San Diego, CA).

Recombinant Human TMPRSS2 protein expressed from yeast (human TMPRSS2 residues 106492, N-terminal 6x His-tag) (cat. # CSB-YP023924HU) was acquired from Cusabio. The fluorogenic peptide substrate, Boc-QAR-AMC·HCl was obtained from Bachem (cat. # I-1550).

## Analysis of cell viability

Sub-confluent Calu-3 cells were incubated for 24 h in the presence of different concentrations (4-fold serial dilutions, 50,000–0.76 nM) of BMS-262084 or camostat. Cells incubated with a medium containing DMSO (solvent) served as controls. Cell viability was assessed using the CellTiter-Glo® Luminescent Cell Viability Assay (Promega) according to the manufacturer's instructions. Data normalization was performed as follows: cell viability in the absence of inhibitor (DMSO-only samples) was set as 100% and the relative viability of cells incubated with the respective inhibitor concentrations was calculated.

## Inhibition of live SARS-CoV-2 cell entry

All infection studies with authentic SARS-CoV-2 were conducted under BSL-3 conditions at the German Primate Center. For stock preparation virus was propagated on Calu-3 (kindly provided by Stephan Ludwig) or Vero E6-TMPRSS2 cells (kindly provided by Stuart Turville) and to ensure that no unwanted S protein mutations occurred during passaging, correct S protein sequences of all SARS-CoV-2 lineages were confirmed by Sanger sequencing for each passage.

BMS-262084- and control-treated Calu-3 cells were infected with SARS-CoV-2 isolate hCoV-19/Germany/FI1103201/2020 (GISAID accession: EPI-ISL_463008; kindly provided by Stephan Ludwig) using a multiplicity of infection (MOI) of 1 and fixed after 24 h. Next, the infected cells were stained with SARS-CoV-2 nucleoprotein-specific (Sino Biologicals, 40143-R019) and Alexa-488-conjugated secondary antibodies (Thermo Fisher Scientific, A-21467) and the infection was analyzed by fluorescence microscopy (nuclei of the cells were stained with DAPI). The same experiment was run with a GFP-expressing vesicular stomatitis virus (VSV, MOI = 0.1)[52], which does not depend on TMPRSS2 for cell entry and is hence not affected by BMS-262084. Finally, the relative infectivity was quantified (normalized to samples without inhibitor = 100% infection) based on the fluorescence intensities in the green channel of the microscopic images, using the ImageJ software[53].

Confirmatory experiments were conducted with BMS-262084 and camostat using plaque assay for virus quantification. A pre-Omicron variant (Delta, AY.1; obtained from Andrew S. Pekosz through BEI Resources, NIAID, NIH; Catalog No. NR-55691) and a recent Omicron sub-lineage (KP.3.1.1) were included. Calu-3 cells were pretreated with different concentrations of BMS-262084 or camostat (diluted in culture medium) or mock-treated. After an incubation period of 2 h at 37 °C, the culture supernatant was aspirated and cells were inoculated 5000 plaque forming units of SARS-CoV-2 Delta or Omicron variants or VSV (diluted in medium containing the respective compound concentration). After an incubation period of 1 h at 37 °C, the inoculum was removed and cells were washed two-times with PBS before they received medium containing fresh compound at the desired concentration. At 48 h post-inoculation, culture supernatants were collected and viral titers determined. For the determination of viral titers, plaque titration was employed using the following protocol. Vero E6-TMPRSS2 cells were seeded in 48-well plates and incubated until they reached confluence. Then, the medium was aspirated and cells were incubated for 1 h with ten-fold serial dilutions of supernatant. Next, the inoculum was removed and cells were washed two-times with PBS before overlay medium (culture medium containing 1% w/v methylcellulose; Sigma-Aldrich, Catalog No. M0512) was added. The cells were incubated for 36 h (VSV), 72 h (SARS-CoV-2 Delta variant AY.1) or 96 h (SARS-CoV-2 Omicron variant KP.3.1.1) until plaques were formed. For plaque counting, the overlay medium was aspirated and cells were washed two-times with PBS and fixed with 4% paraformaldehyde solution (ROTI Histofix, Carl Roth; Catalog No. P087.5) for 1 h at room temperature. Following removal of the paraformaldehyde solution, the cells were stained with crystal violet solution (0.5% crystal violet w/v, 20% ethanol [96%], 79.5% deionized water) for 30 min at room temperature, washed three-times with PBS, air-dried and analyzed using ZEISS Axio Vert. A1 light microscope (equipped with ZEISS A-Plan 2.5×/0.06 M27 objective).

## Inhibition of pseudovirus cell entry

Pseudoviruses bearing VSV-G (control) or the S protein of human coronavirus (HCoV) NL63[54], HCoV-229E[54], SARS-CoV-1[55], MERS-CoV[56], or SARS-CoV-2 lineages B.1 (early pandemic)[57], B.1.617.2 (Delta variant)[58], EG.5.1 (XBB-sublineage, currently circulating)[59], or BA.2.86 (Omicron subvariant)[25] were produced according to a published protocol[60] and inoculated onto Calu-3 cells, which were preincubated (1 h, 37 °C) with different concentrations (10-fold serial dilutions, 25,000–0.25 nM) of BMS-262084 or camostat. Cells incubated with a medium containing DMSO (solvent) served as controls. At 16–18 h postinoculation, pseudovirus cell entry was analyzed by measurement of the activity of virus-encoded firefly luciferase in cell lysates. Data normalization was performed as follows: cell entry in the absence of inhibitor (DMSO-only samples) was set as 0% inhibition and the relative inhibition of cell entry by the respective inhibitor concentrations was calculated.

## Dose-response curve fitting

To determine the inhibitor concentration that causes 50% inhibition of TMPRSS2 activity or (pseudo)virus entry (IC50), we utilize the four-parameter log-logistic model with variable slope. The model is characterized by the following formula:

$$f(x, (b,c,d,e)) = c + \frac{d-c}{1 + \exp(b \cdot (\ln(x) - \ln(e)))} \qquad (2)$$

where $x$ is the concentration of the inhibitor, $b$ is the Hill slope, $c$ is the lower limit (set to 0), $d$ is the upper limit (set to 100) and $e$ is the IC50 value. We employed the curve fitting algorithm available in the SciPy package (version 1.10.1)[61] to derive the Hill slope and the IC50 value, along with their corresponding error estimates.

## Pre-incubation time vs IC50 modeling

To model the relationship between the pre-incubation time and the IC50, we use the following equation inspired by the Morse potential:

$$f(t, (a,D,t_e)) = D \cdot \left(1 - \exp(-a \cdot (t - t_e))\right)^2 \qquad (3)$$

where $t$ is the time, $a$ is the width of the well, $D$ is the IC50 at infinite time (depth of the well) and $t_e$ is the time at which the IC50 reaches its minimum. Again, we made use of SciPy's curve fitting method to obtain parameters $a$, $D$ and $t_e$, with their respective error estimates.

## Reporting summary

Further information on research design is available in the Nature Portfolio Reporting Summary linked to this article.

# Data availability

The molecular simulations and the associated scores generated in this study have been deposited in the Zenodo database under accession code 15508214[62]. Additional computational and experimental data generated in this study are provided in the Supplementary Information file. Source data are provided with this paper.

# Code availability

Our protocol is open-sourced under the MIT license at https://github.com/noegroup/tmprss2_structures/tree/master/scripts[63].

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

## Acknowledgements

K.E. and T.H. acknowledge the financial support of the Deutsche Forschungsgemeinschaft (DFG, German Research Foundation) - SFB 1114, project C03 and SFB/TRR 186, project A12. F.N. acknowledges funding by the European Commission (ERC CoG 772230 "ScaleCell"), the Berlin Mathematics Research Center MATH+ (AA1-10) and the Berlin Institute for the Foundations of Learning and Data (BIFOLD). L.R. acknowledges funding by the European Union's Horizon 2020 research and innovation programme under the Marie Sklodowska-Curie grant agreement no. 897414. R.W. and T.L. acknowledge Bayer AG's PhD scholarships. This work was also supported by the National Center for Advancing Translational Sciences, Division of Preclinical Innovation. S.P. acknowledges funding by the EU project UNDINE (grant agreement number 101057100), the COVID-19-Research Network Lower Saxony (COFONI) through funding from the Ministry of Science and Culture of Lower Saxony in Germany (14-76103-184, projects 7FF22, 6FF22, 10FF22) and the German Research Foundation (Deutsche Forschungsgemeinschaft, DFG; PO 716/11-1). We thank Simon Olsson (Chalmers University) and Moritz Hoffmann (Ethereum Foundation). The following reagent was obtained through BEI Resources, NIAID, NIH: SARS-Related Coronavirus 2, Isolate hCoV-19/USA/CA-VRLC086/2021 (Delta Variant), NR-55691, contributed by Andrew S. Pekosz.

## Author contributions

F.N., M.D.H., M.H. and S.P. designed research. K.E., T.H., L.R. and F.N. conducted & analyzed computational experiments. J.H.S. and M.D.H. conducted TMPRSS2 biochemical assay experiments. N.M., C.R., S.P. and M.H. conducted cell entry experiments. R.W. and T.L. conducted preliminary active learning experiments. K.E., T.H., J.H.S., N.M., L.R., C.R., S.P., M.H., M.D.H. and F.N. analyzed experimental results. K.E., T.H., M.H., L.R. and F.N. wrote the manuscript.

## Funding

## Competing interests

The authors declare no competing interests.
