## [Transparent Peer Review file · Nature Communications]

Simulations and active learning enable efficient identification of an experimentally-validated broad coronavirus inhibitor

Corresponding Author: Professor Frank Noé

Version 0:

Reviewer comments:

Reviewer #1

(Remarks to the Author)

The article "Simulations and active learning enable efficient identification of an experimentally-validated broad coronavirus inhibitor" shows the identification of novel TMPRSS2 inhibitors, using the newly developed active learning approach. They identified a novel and potent TMPRSS2 inhibitor, BMS-262084, which blocks entry of various SARS-CoV-2 variants and other coronaviruses. But the experimental data presented are insufficient to support the claims the authors made, as explained below.

Major

1. If the main point of the manuscript is to show the superior capacity of the h-score over the docking score in identifying more potent inhibitors, as the manuscript claims, the authors should demonstrate its general utility by showing it for more than one target protein.
2. DB03417, the best compound identified from the virtual screening the authors conducted, shows a higher h-score (1.443) than its parental compound BMS-262084 (1.249). It is unclear why the authors evaluated only BMS-262084, not DB03417, in their biochemical and virus inhibition assays.
3. Since the only experiment with authentic virus is image-based quantification of SARS-CoV-2 N protein expression, I would recommend a plaque assay to confirm that BMS-262084 is a potent TMPRSS2 inhibitor.

Minor

1. The information about authentic virus is lacking, such as how it was propagated and titrated. If the authors propagated SARS-CoV-2 in Vero cells, the furin-cleavage site must have been mutated within a few passages (this is well-established fact), and it could have affected the outcome of neutralization assays.
2. In Fig. 4, all Y-axis labels should be improved. Since a and b graphs are derived from the enzymatic TMPRSS2 assay, I would suggest that Y-axis labels should be TMPRSS2 activity inhibition (%) and TMPRSS2 activity IC50, respectively. For c, SARS-CoV-2 entry inhibition. For d-k, PV entry inhibition.
3. In Fig. 4c, the authors should include Camostat as a comparand as they did in other assays.
4. Line 234, subtitle should be "BMS-262084 blocks coronavirus entry into Calu-3 cells".

(Remarks on code availability)

Reviewer #2

(Remarks to the Author)

The paper by Elez et al. reports on a new active learning approach that, combined with MD, significantly reduces the time and computational resources required to identify effective inhibitors of the TMPRSS2 protease. This is a key protein for entering various (corona)viruses into host cells. The authors identify a nanomolar inhibitor, BMS-262084, that shows potent inhibition of TMPRSS2 and the entry of SARS-CoV-2 and other coronaviruses into human lung cells. While the approach seems quite promising in pinpointing potential drug candidates, the methodological character of the studies makes it more suitable for a technical journal. At the same time, a wider application can be considered in further implementations.

Indeed, the method identified a beta-lactam as a covalent inhibitor of a serine protease. The authors claim the screening was successful, as the inhibitor was a nanomolar binder of the pharmacological target. It is well known within the drug discovery community that covalent drugs are usually more potent than non-covalent ones, as a single covalent bond brings orders of magnitude more enthalpic gain relative to non-covalent interactions. Additionally, it has been reported that some beta-lactams can covalently bind serine proteases. Conventional covalent docking tools or even "similarity search engines" would probably have led to similar outcomes. What is really missing in innovative computational drug discovery is the capability to point to completely novel scaffolds/ideas with actual "chemical out-of-the-box thinking."

Other points:

1. While the authors provide an effective framework for TMPRSS2 inhibition, how well this approach generalizes to other targets, particularly those that do not rely on the same structural properties (e.g., serine proteases), remains to be seen. This may be the next step where comparison with the plethora of other similar/different approaches should be considered. What is the real added value? The present study shows that the authors have discovered beta-lactam as a covalent inhibitor of serine protease.

2. Although the paper successfully identifies and validates BMS-262084, the experimental data are somewhat limited in scope. A broader range of in vivo studies or tests on other viral strains would strengthen the impact of the work.

3. There is no mention of hyperparameters or architectural specifics for the k-means algorithm, the SVM regressor, or the autoencoder (if it is trained from scratch). Including these details is crucial for reproducibility and understanding the model's performance.

4. The paper does not compare the performance of the active learning-based pipeline and conventional covalent docking tools. A direct comparison would be helpful to quantify the advantages of the proposed approach, particularly in identifying covalent inhibitors.

5. The paper does not clarify whether the autoencoder used in this pipeline is pretrained or trained from scratch. This is an important detail that should be explicitly stated to provide clarity on the model's training process.

6. While the authors provide a GitHub repository containing scripts for the MD part, no code concerning the active learning models or the pipeline for candidate selection is available. Making the complete code available would be beneficial to facilitate transparency and reproducibility.

7. The CDDD (Continuous Data-Driven Descriptors) encodings are referenced in the paper but are not clearly defined. Please explicitly define CDDD when introducing the autoencoder to avoid confusion and ensure readers fully understand its role (the abbreviation "CDDD" should be defined upon first mention, especially when introducing the autoencoder).

In conclusion, the idea can be interesting to help modern drug discovery, which is struggling with appropriate tools to predict ligand affinity, particularly for innovative and non-conventional drugs. Physics-based approaches have strived for years to compute binding free energy accurately (error less than 1 kcal/mol). A routine methodology to address this issue is still missing. Neither physical modeling nor machine learning has provided real breakthroughs in the field. Notwithstanding, the present methodology seems promising, it is, for the moment, a fascinating technical evolution. The present reviewer does not grasp how much this new approach can help address the significant challenges of computational drug discovery.

(Remarks on code availability)

Reviewer #3

(Remarks to the Author)

In this work, the authors present a methodology integrating molecular dynamics and active learning to expedite the process of virtual screening, claiming the successful identification of a potent TMPRSS2 inhibitor as a testament to the method's efficacy. Though the performance is impressive, there are still important questions that need to be further addressed.

1. The Novelty of this work necessitates more emphasis in order to meet the standards expected by a journal like Nat Commun:

1.1.) The methodology primarily amalgamates well-established techniques—target-specific scoring functions, molecular dynamics simulations, and active learning—without introducing innovative algorithms or models. This combination, while potentially useful for specific targets shown by the authors, seemingly does not meet the high novelty from the perspective of the protocol. More importantly, the static h-score, tailored specifically for TMPRSS2, underscores a significant limitation in the broader applicability of the proposed method. The absence of a generalized approach for creating target-specific scoring functions for diverse targets may further constrain its utility.

1.2.) The accuracy ranking (dynamic h-score > static h-score > docking score) aligns predictably with established literature (molecular dynamics > target-specific scoring functions > generic scoring functions) (<https://doi.org/10.1093/bib/bbaa410>), suggesting a lack of fresh insights from the study.

1.3. The claimed acceleration in virtual screening primarily derives from the use of an active learning algorithm, a technique already established in the field (<https://pubs.rsc.org/en/content/articlehtml/2021/sc/d0sc06805e>) (<https://pubs.rsc.org/en/content/articlehtml/2024/dd/d3dd00227f>) (<https://pubs.acs.org/doi/10.1021/acs.jcim.2c00554>), thus not sufficiently contributing novel methodology.

2. Practical Relevance: The size of the ligand library utilized in this study is constrained, a limitation not typically encountered in actual drug discovery scenarios. This constraint may detract from the persuasiveness of the effectiveness demonstrated. For instance, as detailed in Table 1, the authors evaluated various pipeline configurations using the DrugBank database, which contains merely 8,918 compounds. This figure starkly contrasts with the expansive compound libraries typically employed in real-world drug discovery efforts. Furthermore, the inclusion of only four known TMPRSS2 inhibitors within the library for evaluation purposes—specifically for calculating Spearman's rank correlation coefficient across different pipeline configurations—further undermines the robustness of the results. Such a limited dataset may not provide a sufficiently rigorous test of the pipeline's efficacy.

3. Related questions that might be interesting to readership:

3.1.) Active learning can indeed enhance the efficiency of discovering active molecules. However, during the active learning cycle, it is unavoidable to use machine learning models to rank the potential activity values of molecules in the library, which has natural bias that they often favor compounds that better match the distribution of the training set. Does this imply that while active learning improves the efficiency of discovering active molecules, it may also reduce the likelihood of identifying structures with entirely new scaffolds? I believe this issue can be addressed by comparing the diversity of molecular scaffolds selected with and without an active learning approach.

3.2.) During the computation of the h-score, were the docking conformations of all the protein structures generated by MD simulations considered, or was only one used? I could not easily find relevant information on this in the manuscript.

3.3.) It seems that both holo and apo structures were employed when generating protein conformations using MD simulations. However, the enrichment effect based on holo structures is generally much better than that based on apo structures in standard virtual screening. Does this phenomenon still hold true under the framework you proposed? Could some related investigations be conducted?

4. Minor points: the existing literature is replete with nanomolar inhibitors of TMPRSS2, as exemplified by the study published in Nature (<https://www.nature.com/articles/s41586-022-04661-w>). This should be cited.

(Remarks on code availability)

Version 1:

Reviewer comments:

Reviewer #1

(Remarks to the Author)

The authors addressed most of my concerns, and the revised manuscript has been significantly improved. One thing that should be clarified is whether the authors performed plaque assays in Fig 4c and 4d as they claim. If they did, it would be more appropriate to present the data with the Y-axis in PFU/ml, and the plaque assay method should be described in the Methods section.

(Remarks on code availability)

Reviewer #2

(Remarks to the Author)

The authors have only partially addressed the points I raised. The overall novelty of the study/method is not completely clear to me, particularly when compared to other, more established approaches for drug design. Covalent inhibitors can be identified in many different manners, and well-known to be very potent enzyme inhibitors. In conclusion, it is an interesting study, still not clear the level of novelty/impact to justify publication in Nature Comm. The Editor has a much wider view of the study and all the referees' reports to provide a final assessment.

(Remarks on code availability)

Reviewer #3

(Remarks to the Author)

All my comments have been addressed appropriately. I still tend to believe that this work is not on the "Nature level" of novelty and significance. However, I think this is mainly an editorial problem and on my behalf, the paper is now acceptable for publication.

(Remarks on code availability)

Reviewer #1 (Remarks to the Author):

The article “Simulations and active learning enable efficient identification of an experimentally-validated broad coronavirus inhibitor” shows the identification of novel TMPRSS2 inhibitors, using the newly developed active learning approach. They identified a novel and potent TMPRSS2 inhibitor, BMS-262084, which blocks entry of various SARS-CoV-2 variants and other coronaviruses. But the experimental data presented are insufficient to support the claims the authors made, as explained below.

Major

1. If the main point of the manuscript is to show the superior capacity of the h-score over the docking score in identifying more potent inhibitors, as the manuscript claims, the authors should demonstrate its general utility by showing it for more than one target protein.

We thank the reviewer for the suggestion to evaluate the general utility of the h-score across multiple target proteins. To demonstrate the broader applicability of our approach, we extended our methodology to a set of trypsin-domain proteins by learning a trypsin-domain-specific score from experimental binding affinity data. This learned score, based on the same observables as the h-score, achieves a strong correlation (0.80) with experimental binding affinities, demonstrating that the underlying principles of the h-score generalize beyond a single target. For drug discovery in other Trypsin proteases, we recommend using this score. Our new results have been incorporated into Figure 2 and described in the new results section "Learned score generalizes to trypsin-domain proteins" (please see lines 121–151).

Please note that the computational and experimental assays in the paper are still focused on TMPRSS2 and employ the TMPRSS2-specific score, as rerunning all computational and experimental assays with a new score would be an entirely new study.

2. DB03417, the best compound identified from the virtual screening the authors conducted, shows a higher h-score (1.443) than its parental compound BMS-262084 (1.249). It is unclear why the authors evaluated only BMS-262084, not DB03417, in their biochemical and virus inhibition assays.

We appreciate the reviewer’s observation regarding our choice of compound for biochemical and virus inhibition assays. While DB03417 showed a higher h-score than its parental compound BMS-262084 in our virtual screening, we could not evaluate DB03417 experimentally as we could not acquire the compound or a synthesis service that could produce it.

Furthermore, DB03417 was likely derived from the PDB structure 1RXP in which BMS-262084 was captured covalently bound to trypsin. Structurally, DB03417 represents a

hydrolyzed variant of BMS-262084 that resulted from incorrect processing of the PDB structure, in which the ligand was simply detached from the covalent complex without accounting for the hydrolysis step (see Figure 5 and Figure S1). In contrast, BMS-262084 is synthetically accessible and biologically relevant. Given these considerations, we prioritized BMS-262084 for experimental evaluation.

3. Since the only experiment with authentic virus is image-based quantification of SARS-CoV-2 N protein expression, I would recommend a plaque assay to confirm that BMS-262084 is a potent TMPRSS2 inhibitor.

As suggested, we conducted confirmatory experiments with BMS-262084 and camostat using plaque assay for virus quantification. Further, we included a pre-Omicron variant (Delta, AY.1) and a recent Omicron sublineage (KP.3.1.1) for direct comparison. The new results confirm the data obtained by image-based quantification, indicating that BMS-262084 is a potent inhibitor of SARS-CoV-2 infection of Calu-3 lung cells. The data have been incorporated into Figure 4 and described in the results section “BMS-262084 blocks coronavirus entry into Calu-3 cells” (please see lines 278-285).

Minor

1. The information about authentic virus is lacking, such as how it was propagated and titrated. If the authors propagated SARS-CoV-2 in Vero cells, the furin-cleavage site must have been mutated within a few passages (this is well-established fact), and it could have affected the outcome of neutralization assays.

The SARS-CoV-2 isolates used for the live-virus inhibition experiments were propagated on Calu-3 (B.1.1) or Vero E6-TMPRSS2 (AY.1, KP.3.1.1) cells and titrated on Vero E6-TMPRSS2 cells. This information has been added to the methods section: “For stock preparation virus was propagated on Calu-3 (kindly provided by Stephan Ludwig) or Vero E6-TMPRSS2 cells (kindly provided by Stuart Turville) and viral titers were determined by plaque titration on Vero E6-TMPRSS2 cells.” (please see lines 684-688). Further, we sequenced the S protein gene of our stock viruses to ensure that the furin cleavage site was intact. No mutations (apart from the ones that are characteristic of the respective isolates) were detected.

2. In Fig. 4, all Y-axis labels should be improved. Since a and b graphs are derived from the enzymatic TMPRSS2 assay, I would suggest that Y-axis labels should be TMPRSS2 activity inhibition (%) and TMPRSS2 activity IC50, respectively. For c, SARS-CoV-2 entry inhibition. For d-k, PV entry inhibition.

As suggested, we modified the y-axis labels of the graphs for clarity.

3. In Fig. 4c, the authors should include Camostat as a comparand as they did in other assays.

As described in our response to Major point #3 (please see above), we conducted additional live-virus inhibition experiments that were quantified by plaque titration and included camostat for direct comparison with BMS-262084. The results confirm our initial live-virus and pseudovirus data indicating that BMS-262084 is a potent SARS-CoV-2/Coronavirus entry inhibitor and that BMS-262084 inhibits SARS-CoV-2 S protein-driven entry with higher efficiency compared to the well-known TMPRSS2 inhibitor camostat. The data have been incorporated into Figure 4 and described in the results section: “BMS-262084 blocks coronavirus entry into Calu-3 cells” (please see lines 278-285).

4. Line 234, subtitle should be “BMS-262084 blocks coronavirus entry into Calu-3 cells”.

We modified the subtitle as suggested.

Reviewer #2 (Remarks to the Author):

The paper by Elez et al. reports on a new active learning approach that, combined with MD, significantly reduces the time and computational resources required to identify effective inhibitors of the TMPRSS2 protease. This is a key protein for entering various (corona)viruses into host cells. The authors identify a nanomolar inhibitor, BMS-262084, that shows potent inhibition of TMPRSS2 and the entry of SARS-CoV-2 and other coronaviruses into human lung cells. While the approach seems quite promising in pinpointing potential drug candidates, the methodological character of the studies makes it more suitable for a technical journal. At the same time, a wider application can be considered in further implementations.

Indeed, the method identified a beta-lactam as a covalent inhibitor of a serine protease. The authors claim the screening was successful, as the inhibitor was a nanomolar binder of the pharmacological target. It is well known within the drug discovery community that covalent drugs are usually more potent than non-covalent ones, as a single covalent bond brings orders of magnitude more enthalpic gain relative to non-covalent interactions. Additionally, it has been reported that some beta-lactams can covalently bind serine proteases. Conventional covalent docking tools or even “similarity search engines” would probably have led to similar outcomes. What is really missing in innovative computational drug discovery is the capability to point to completely novel scaffolds/ideas with actual “chemical out-of-the-box thinking.”

Other points:

1. While the authors provide an effective framework for TMPRSS2 inhibition, how well this approach generalizes to other targets, particularly those that do not rely on the same structural properties (e.g., serine proteases), remains to be seen. This may be the next step where comparison with the plethora of other similar/different approaches should be considered. What is the real added value? The present study shows that the authors have discovered beta-lactam as a covalent inhibitor of serine protease.

We appreciate the reviewer's comments regarding the generalizability of our approach beyond TMPRSS2 and serine proteases. In the revised manuscript, we demonstrate that the h-score, which was initially developed for TMPRSS2, captures key molecular interactions relevant to inhibitor potency and can be extended beyond a single target. We expanded our analysis to a broader set of trypsin-domain proteins, learning a trypsin-domain-specific score from experimental binding affinity data. As described in the new results section "Learned score generalizes to trypsin-domain proteins" (lines 121–151), this learned score achieves a strong correlation (0.80) with experimental

binding affinities, supporting the broader applicability of our approach within this protein family.

The added value of our study is providing a pathway for translating insights from biochemical mechanisms to a score-based description and subsequent hit discovery that is experimentally validated and we show applies to a broad class of enzymes. Please note that the experimental and computational assays still focus on TMPRSS2 and use the TMPRSS2-specific score, as rerunning the entire experimental and computational pipeline with a new score would be an entirely new study. We acknowledge that the generalization of our method to non-serine proteases or structurally distinct target classes remains an important direction for future research.

2. Although the paper successfully identifies and validates BMS-262084, the experimental data are somewhat limited in scope. A broader range of in vivo studies or tests on other viral strains would strengthen the impact of the work.

This study focuses on the bioinformatics pipeline for the identification of broad coronavirus inhibitors by targeting TMPRSS2, while the experimental part represents a validation of this strategy by proofing that the identified compound has strong antiviral activity. This validation includes a variety of in vitro assays that directly analyze the impact of the compound on TMPRSS2 enzyme activity, spike protein-driven cell entry in the context of pseudoviruses and Calu-3 lung cell infection by authentic SARS-CoV-2 isolates. Furthermore, our initial submission contains data on a total of ten viruses: two authentic viruses, VSV and SARS-CoV-2 (B.1.1 lineage), as well as eight different pseudoviruses bearing spike proteins of four different SARS-CoV-2 lineages (B.1, B.1.617.2, EG.5.1, BA.2.86), human coronavirus NL63, human coronavirus 229E, SARS-CoV-1, or MERS-CoV.

In the revised manuscript we have now additionally included new live-virus data in which two additional SARS-CoV-2 isolates, an early SARS-CoV-2 variant (Delta, AY.1) and a recent Omicron sublineage (KP.3.1.1), were analyzed (please see Figure 4). Thus, a total of twelve different viruses (four authentic viruses and eight pseudoviruses) have been analyzed, which we feel is sufficient to prove that the identified compound is a potent inhibitor of coronavirus cell entry. The addition of in vivo studies is beyond the scope of the present study.

3. There is no mention of hyperparameters or architectural specifics for the k-means algorithm, the SVM regressor, or the autoencoder (if it is trained from scratch). Including these details is crucial for reproducibility and understanding the model's performance.

We thank the reviewer for pointing this out. These are the specifics of the algorithms:

- K-means was used with the following non-default parameters: 20 initializations and a maximum of 400 iterations.
- SVM regressor was used with the default settings in scikit-learn.

- Autoencoder was pretrained and not trained from scratch, so no new hyperparameters or architectural adjustments were introduced.

We have included these parameter details in the revised manuscript (see lines 598, 614-615 and 622-623).

4. The paper does not compare the performance of the active learning-based pipeline and conventional covalent docking tools. A direct comparison would be helpful to quantify the advantages of the proposed approach, particularly in identifying covalent inhibitors.

We appreciate the reviewer's suggestion to compare our approach with conventional covalent docking tools. However, our method is designed to provide a unified scoring framework applicable to both covalent and noncovalent inhibitors, without bias toward either. The goal is to offer a versatile scoring system that can evaluate both types of inhibitors together. Additionally, covalent docking typically requires a more complex setup to accurately model covalent interactions, whereas our approach simplifies this process, offering an effective alternative for assessing binding interactions in a broader context. Given that our focus is on developing a scoring framework that integrates both inhibitor types, a fair comparison with covalent docking methods cannot be achieved.

5. The paper does not clarify whether the autoencoder used in this pipeline is pretrained or trained from scratch. This is an important detail that should be explicitly stated to provide clarity on the model's training process.

We apologize for the lack of clarity regarding the training process of the autoencoder. To clarify, the autoencoder used in our pipeline is pretrained based on the methodology reported in the previous study by Winter et al. We updated the manuscript to explicitly state that the autoencoder was pretrained (please see line 598).

6. While the authors provide a GitHub repository containing scripts for the MD part, no code concerning the active learning models or the pipeline for candidate selection is available. Making the complete code available would be beneficial to facilitate transparency and reproducibility.

We thank the reviewer for the suggestion regarding code availability. We added the code for the active learning models and the candidate selection pipeline to the GitHub repository (https://github.com/noegroup/tmprss2_structures/tree/master/scripts).

7. The CDDD (Continuous Data-Driven Descriptors) encodings are referenced in the paper but are not clearly defined. Please explicitly define CDDD when introducing the autoencoder to avoid confusion and ensure readers fully understand its role (the abbreviation "CDDD" should be defined upon first mention, especially when introducing the autoencoder).

We thank the reviewer for this observation. We have made the change as you recommended and explicitly defined CDDD (continuous data-driven descriptors) upon first mention (please see lines 600-601).

In conclusion, the idea can be interesting to help modern drug discovery, which is struggling with appropriate tools to predict ligand affinity, particularly for innovative and non-conventional drugs. Physics-based approaches have strived for years to compute binding free energy accurately (error less than 1 kcal/mol). A routine methodology to address this issue is still missing. Neither physical modeling nor machine learning has provided real breakthroughs in the field. Notwithstanding, the present methodology seems promising, it is, for the moment, a fascinating technical evolution. The present reviewer does not grasp how much this new approach can help address the significant challenges of computational drug discovery.

We thank the reviewer for this positive outlook. We share the vision that physics-based model can add important value in the drug discovery and development pipeline when combined with high-throughput tools. Certainly there are a lot more important work to be done in the future.

Reviewer #3 (Remarks to the Author):

In this work, the authors present a methodology integrating molecular dynamics and active learning to expedite the process of virtual screening, claiming the successful identification of a potent TMPRSS2 inhibitor as a testament to the method's efficacy. Though the performance is impressive, there are still important questions that need to be further addressed.

1. The Novelty of this work necessitates more emphasis in order to meet the standards expected by a journal like Nat Commun:

1.1.) The methodology primarily amalgamates well-established techniques—target-specific scoring functions, molecular dynamics simulations, and active learning—without introducing innovative algorithms or models. This combination, while potentially useful for specific targets shown by the authors, seemingly does not meet the high novelty from the perspective of the protocol. More importantly, the static h-score, tailored specifically for TMPRSS2, underscores a significant limitation in the broader applicability of the proposed method. The absence of a generalized approach for creating target-specific scoring functions for diverse targets may further constrain its utility.

We thank the reviewer for the feedback regarding the novelty and broader applicability of our methodology. While it is true that our approach builds on well-established techniques, the key contribution of this work lies in the systematic integration of these techniques to develop an effective and interpretable scoring function for inhibitor potency prediction. Our study demonstrates how carefully designed observables - rooted in the molecular mechanism of action - can improve the predictive power of scoring functions.

Regarding the generalization of our method, we acknowledge that the static h-score was initially developed for TMPRSS2, which may raise concerns about its broader applicability. However, to address this limitation, we have taken a data-driven approach and demonstrated that the same observables can be used to learn a generalizable scoring function for trypsin-domain proteins. As described in the new results section "Learned score generalizes to trypsin-domain proteins" (please see lines 121–151), the learned score achieves a strong correlation (0.80) with experimental binding affinities across multiple trypsin-domain proteins, indicating that the fundamental principles underlying the h-score extend beyond a single target. Please note that the experimental and computational screens run in this study still focus on TMPRSS2 using the TMPRSS2-specific score as rerunning them with a different score would be an entirely new study. We agree that further generalization to diverse targets is an important next step and we are actively working toward a general protease score that could extend our approach to a broader range of proteases. Additionally, our methodology provides a blueprint for

constructing target-specific scoring functions based on mechanistic insights, which could be adapted to diverse protein families in future work.

1.2.) The accuracy ranking (dynamic h-score > static h-score > docking score) aligns predictably with established literature (molecular dynamics > target-specific scoring functions > generic scoring functions) (<https://doi.org/10.1093/bib/bbaa410>), suggesting a lack of fresh insights from the study.

Indeed, this ranking is expected and not the relevant novelty of this study, but there are important insights in the details. In our opinion, two stand out:

1. The target-specific scores vastly outperform generic scores for the cases studied and using them comes at no additional cost. In the revised version, we generalize the score to trypsin-domain proteins, thus providing a more general tool for future studies.
2. MD simulations help, but they are mostly beneficial for incorporating the flexibility of the target, while running MD simulations of protein-ligand complexes for all screened ligands provides a minimal performance gain at a large computational overhead.

1.3. The claimed acceleration in virtual screening primarily derives from the use of an active learning algorithm, a technique already established in the field (<https://pubs.rsc.org/en/content/articlehtml/2021/sc/d0sc06805e>) (<https://pubs.rsc.org/en/content/articlehtml/2024/dd/d3dd00227f>) (<https://pubs.acs.org/doi/10.1021/acs.jcim.2c00554>), thus not sufficiently contributing novel methodology.

We appreciate the reviewer's feedback and recognize that active learning is a well-established technique in virtual screening. However, we don't claim that active learning is the key novelty, nor is it the key driver of performance in our study - active learning is an important infrastructure, but the largest performance boost comes from employing a target-specific score and considering flexibility in the target (see reply to 1.2). This combination also effectively reduces the experimental validation and not only the computational cost, a level of synergy that, to our knowledge, has not been systematically demonstrated in prior studies.

Such active learning frameworks incorporating ML, simulation and experiment are quite complex systems of combined technologies, so that in a successfully system there's usually not a single technology that has never been considered before, but it's rather the careful combination of these parts that gives rise to better performance.

2. Practical Relevance: The size of the ligand library utilized in this study is constrained, a limitation not typically encountered in actual drug discovery scenarios. This constraint may detract from the persuasiveness of the effectiveness demonstrated. For instance, as detailed in Table 1, the authors evaluated various pipeline configurations using the DrugBank database, which contains merely 8,918 compounds. This figure starkly contrasts with the expansive

compound libraries typically employed in real-world drug discovery efforts. Furthermore, the inclusion of only four known Tmprss2 inhibitors within the library for evaluation purposes—specifically for calculating Spearman’s rank correlation coefficient across different pipeline configurations—further undermines the robustness of the results. Such a limited dataset may not provide a sufficiently rigorous test of the pipeline’s efficacy.

Thank you for highlighting the concerns regarding the size of the ligand library used in our study. We would like to clarify that the constrained library size was an intentional decision driven by the specific nature of our approach. In this study, we conducted molecular dynamics (MD) simulations for every compound to evaluate the dynamic target-specific inhibition score. This process is computationally intensive and it is not feasible to systematically apply this to a library of arbitrary size. Therefore, the chosen library serves primarily as a proof of concept, demonstrating the potential of our method to prioritize compounds based on a more accurate scoring metric. While we acknowledge that real-world drug discovery efforts often involve larger libraries, the 8,918 compounds in the DrugBank dataset are still sufficiently representative to support the findings of this study.

Regarding the use of only four known Tmprss2 inhibitors for calculating the Spearman rank correlation coefficient, we recognize that this limited sample size may impact the statistical reliability of this particular metric. In response to this, we have included a note in the revised manuscript to highlight that the Spearman correlation was computed using only four inhibitors (please see the caption of Tab. 1), thereby acknowledging the limitations associated with this. However, we would like to emphasize that other evaluation metrics utilized in the study, such as the reduction in computational and experimental costs, provide a more comprehensive and robust assessment of the methodology. We are confident that these metrics underscore the effectiveness of our approach, even with a limited sample size of known inhibitors.

3. Related questions that might be interesting to readership:

3.1.) Active learning can indeed enhance the efficiency of discovering active molecules. However, during the active learning cycle, it is unavoidable to use machine learning models to rank the potential activity values of molecules in the library, which has natural bias that they often favor compounds that better match the distribution of the training set. Does this imply that while active learning improves the efficiency of discovering active molecules, it may also reduce the likelihood of identifying structures with entirely new scaffolds? I believe this issue can be addressed by comparing the diversity of molecular scaffolds selected with and without an active learning approach.

We appreciate the reviewer’s question regarding the potential bias of active learning toward molecules resembling those in the training set, which could limit scaffold diversity. To investigate this, we compared the scaffold diversity of selected compounds

with and without active learning. Specifically, we ran the pipeline setups described in row 3 of Table 1 (receptor ensemble, dynamic h-score and active learning enabled) and row 6 of Table 1 (receptor ensemble, dynamic h-score, active learning disabled) for an initial round plus nine extension rounds, screening 10% of the database in total. To assess scaffold diversity, we extracted unique Murcko scaffolds from the selected compounds. The similarity of unique scaffolds was evaluated using Tanimoto similarity of their Morgan fingerprints, generated with a radius of 3 and a fingerprint size of 2048 bits. The mean scaffold similarity was calculated by averaging pairwise similarity scores across all selected compounds in each round. The results are summarized in the figure below.

Our analysis shows that while active learning improves efficiency in identifying active molecules, it selects slightly fewer unique scaffolds compared to random selection. However, the mean scaffold similarity remains comparable between the two approaches, indicating that active learning does not overly favor highly similar molecules. These results suggest that although active learning introduces some bias, it still explores a diverse chemical space.

3.2.) During the computation of the h-score, were the docking conformations of all the protein structures generated by MD simulations considered, or was only one used? I could not easily find relevant information on this in the manuscript.

We utilize 20 receptor structures extracted from MD simulations and each ligand is docked to all 20 receptor structures, resulting in 20 protein-ligand conformations (using the highest scoring pose per receptor structure). The static h-score is then computed as the mean of the top 3 protein-ligand conformations. For the dynamic h-score, we calculate the score as the mean of the top 3 protein-ligand trajectories. We updated the manuscript to explicitly include these details for clarity (please see lines 518-522).

3.3.) It seems that both holo and apo structures were employed when generating protein conformations using MD simulations. However, the enrichment effect based on holo structures is generally much better than that based on apo

structures in standard virtual screening. Does this phenomenon still hold true under the framework you proposed? Could some related investigations be conducted?

We thank the reviewer for the insightful question regarding the impact of using apo versus holo structures in our framework. To address this, we conducted an analysis using the same pipeline setup as described in row 2 of Table 1 (receptor ensemble, static h-score and active learning enabled), but with receptor ensembles composed of either only apo or only holo structures. The results are reported in the table below.

Target	Compounds computationally screened	Total simulation time (h)	Compounds to screen experimentally	r_s
Receptor ensemble apo-only	442.8	*1254.6	23.2	0.8
Receptor ensemble holo-only	262.4	*743.5	6.0	0.4

The results indicate that using only holo structures leads to better performance compared to using only apo structures. Specifically, the holo-only setup on average results in fewer compounds being computationally screened and fewer experimental validations needed to identify known inhibitors. This aligns with standard virtual screening observations that holo structures generally provide a more effective starting point for identifying potent inhibitors. However, it is worth noting that the full receptor ensemble (including both apo and holo structures) still slightly outperforms the holo-only setup in terms of reducing the number of compounds to screen experimentally, requiring just 5.6 compounds on average. Additionally, the apo-only ensemble performs reasonably well, ranking known inhibitors among the top 23.2 compounds on average.

4. Minor points: the existing literature is replete with nanomolar inhibitors of TMPRSS2, as exemplified by the study published in Nature (<https://www.nature.com/articles/s41586-022-04661-w>). This should be cited.

We added the citation as suggested (please see line 49).

Reviewer #1 (Remarks to the Author):

The authors addressed most of my concerns, and the revised manuscript has been significantly improved. One thing that should be clarified is whether the authors performed plaque assays in Fig 4c and 4d as they claim. If they did, it would be more appropriate to present the data with the Y-axis in PFU/ml, and the plaque assay method should be described in the Methods section.

As suggested, in Fig. 4c and 4d we now present the data with the Y-axis in PFU/ml. We also added a detailed description of the assay and the plaque titration method to the Methods section (please see paragraph 'Inhibition of live SARS-CoV-2 cell entry').

Reviewer #2 (Remarks to the Author):

The authors have only partially addressed the points I raised. The overall novelty of the study/method is not completely clear to me, particularly when compared to other, more established approaches for drug design. Covalent inhibitors can be identified in many different manners, and well-known to be very potent enzyme inhibitors. In conclusion, it is an interesting study, still not clear the level of novelty/impact to justify publication in Nature Comm. The Editor has a much wider view of the study and all the referees' reports to provide a final assessment.

We thank the reviewer for their comments and for acknowledging the interest of our study. We have made additional clarifications in the revised manuscript to better highlight the novelty and impact of our approach.

Reviewer #3 (Remarks to the Author):

All my comments have been addressed appropriately. I still tend to believe that this work is not on the "Nature level" of novelty and significance. However, I think this is mainly an editorial problem and on my behalf, the paper is now acceptable for publication.

We thank the reviewer for acknowledging our revisions and for recommending the manuscript for publication. We appreciate their feedback throughout the review process.